# A comparison of software for analysis of rare and common short tandem repeat (STR) variation using human genome sequences from clinical and population-based samples

John W. Oketch[1], Louise V. Wain[2,3], Edward J. Hollox[1] *

1 Department of Genetics and Genome Biology, University of Leicester, Leicester, United Kingdom,
2 Department of Population Health Sciences, University of Leicester, Leicester, United Kingdom, 3 National Institute for Health Research, Leicester Respiratory Biomedical Research Centre, Glenfield Hospital, Leicester, United Kingdom

* ejh33@le.ac.uk

**Data Availability Statement:** Code for analyses, and the full set of genotype calls at the clinical and forensic loci are available at https://doi.org/10.

## Abstract

Short tandem repeat (STR) variation is an often overlooked source of variation between genomes. STRs comprise about 3% of the human genome and are highly polymorphic. Some cause Mendelian disease, and others affect gene expression. Their contribution to common disease is not well-understood, but recent software tools designed to genotype STRs using short read sequencing data will help address this. Here, we compare software that genotypes common STRs and rarer STR expansions genome-wide, with the aim of applying them to population-scale genomes. By using the Genome-In-A-Bottle (GIAB) consortium and 1000 Genomes Project short-read sequencing data, we compare performance in terms of sequence length, depth, computing resources needed, genotyping accuracy and number of STRs genotyped. To ensure broad applicability of our findings, we also measure genotyping performance against a set of genomes from clinical samples with known STR expansions, and a set of STRs commonly used for forensic identification. We find that HipSTR, ExpansionHunter and GangSTR perform well in genotyping common STRs, including the CODIS 13 core STRs used for forensic analysis. GangSTR and Expansion-Hunter outperform HipSTR for genotyping call rate and memory usage. ExpansionHunter denovo (EHdn), STRling and GangSTR outperformed STRetch for detecting expanded STRs, and EHdn and STRling used considerably less processor time compared to GangSTR. Analysis on shared genomic sequence data provided by the GIAB consortium allows future performance comparisons of new software approaches on a common set of data, facilitating comparisons and allowing researchers to choose the best software that fulfils their needs.

## Introduction

Over the past two decades, most research on the contribution of genetic variation in the human genome to disease has focused on single nucleotide variation. Short tandem repeat

25392/leicester.data.22041020. Genotype call vcf files for GangSTR and HipSTR and ExpansionHunter are available for the Genome In a bottle samples are at https://ftp-trace.ncbi.nlm.nih. gov/ReferenceSamples/giab/data/ and https://doi. org/10.25392/leicester.data.22041020 Genotype call vcf files for GangSTR, ExpansionHunter and HipSTR are available for the 1000 Genomes samples used are at https://doi.org/10.25392/ leicester.data.22041020.

**Funding:** "JWO is funded by a Wellcome Trust PhD studentship as part of the Wellcome Trust Genetic Epidemiology and Public Health Genomics Doctoral Training Programme by grant number 218505/Z/19/Z. LWV holds a GSK/Asthma+Lung UK Chair in Respiratory Research (C17-1). The research was partially supported by the National Institute for Health Research (NIHR) Leicester Biomedical Research Centre. There was no additional external funding received for this study.

**Competing interests:** The authors have declared that no competing interests exist.

(STR) variation has been generally overlooked, as it is not readily assayed by chip-based hybridisation approaches. STRs are short DNA sequence motifs, typically 2-6bp in size, that are repeated multiple times in tandem. They are distributed throughout the human genome, comprising about 3% of the genome sequence, and have a mutation rate generally much higher than single nucleotide variants (around $2x10^{-3}$ per locus per generation for STRs compared to $10^{-8}$ for single nucleotide variants) [1–4]. They are frequently polymorphic with multiple alleles in a population, as expected by neutral population genetics theory for loci with high mutation rates.

Particular STRs cause a variety of severe rare monogenic diseases that are inherited in a Mendelian manner [5]. These include triplet-repeat expansion diseases, such as Huntington's disease and myotonic dystrophy, caused by expansion of 3 bp repeat STRs in the coding region of *HTT*, and *DMPK* genes respectively. Some disorders also occur with other repeat motifs in both coding and in non-coding regions [5–8]. Emerging evidence shows that STRs modulate disease risk in several complex diseases such as autism and neurodegenerative disorders [9, 10].

STRs can affect the function of genes in three main ways, either by directly altering the coding region of genes, by disrupting introns, or by affecting expression levels as part of enhancer elements. For example, in addition to STRs encoding coding CAG-repeats leading to polyglutamine tracts in genes such as *HTT*, polymorphic STRs with GCA, GCC, GCG or GCT motifs also encode polyalanine tracts of varying length in a variety of genes [11]. Other STRs can disrupt introns, such as the expansion of an intronic ATTCT-repeat in spinocerebellar ataxia type 10 [12]. It is well-established that STR variation is associated with expression levels of genes both at a single gene level and at a genome-wide level [13–15]. It is unclear the extent to which STRs are responsible for genomewide association signals detected using SNPs. Linkage disequilibrium between STRs and SNPs varies, but is expected to be lower than between two SNPs at the same recombination distance because the high mutation rate of STRs rapidly breaks down linkage disequilibrium over time. This is shown in attempts to impute STR length genotypes from SNP genotypes, as biallelic STRs can generally be reliably imputed but multiallelic STRs are much less reliably imputed [16].

Given the potential importance of STRs in human disease, genome-wide studies examining STR variation are clearly warranted. The first generation of STR-calling software, including LobSTR, RepeatSeq and HipSTR [17–19], aimed to genotype STRs using sequencing reads that completely span the STR. This led to the limitation that STRs longer than the sequencing read length could not be genotyped. The second generation of STR-calling software uses information from the distance between the paired sequencing reads, in addition to direct sequence information across the repeat, to genotype the STR. This attempts to capture information on STRs that are longer than the sequencing read length and is particularly effective at identifying large expansions at an STR [20].

This study aimed to compare software, suitable for biobank-scale data analysis, that genotypes common STRs and identifies rarer STR expansions genome-wide on short-read genome sequences. This will inform approaches to investigate the role of genomic STR variants in polygenic human diseases. We selected software tools that had at least one of the following characteristics: not limited by read length, able to estimate STR allele length, or had not been compared previously. Three recent pieces of software are focused on profiling large repeat expansions: Expansion Hunter De Novo (EHdn) [21], STRetch [22] and STRling [23]. They all use sequence reads that span STRs and mate pair distance information to identify large expansions, however, unlike STRetch, EHdn and STRling do not require a predefined catalogue of STRs to genotype an STR locus. Therefore EHdn and STRling can also identify novel STR loci not assembled in the reference genome. STRetch, STRling and EHdn can all detect and genotype large repeat expansions in a sample that are outliers from the population distribution of

allele lengths. Here we assess the ability of STRetch, STRling and EHdn to identity rarer STR expansions by running them on 116 PCR-free short-read whole genome seqeunces containing clinically-validated trinucleotide repeat expansions, normal repeat sizes and or pre-mutations [24].

We also compare three software tools that focus on genome-wide genotyping of both short and expanded (longer than the sequencing read length) repeat arrays given a reference catalogue of STR genomic coordinates: ExpansionHunter, GangSTR and HipSTR. ExpansionHunter and GangSTR use short-read sequence mate-pair distance information together with STR-spanning sequence reads [24, 25]. Although HipSTR does not use mate-pair distance information and therefore limited by sequence read length, a previous study showed that it outperformed its counterparts [19], and therefore was included in this study. These three tools report diploid allele lengths. In addition, HipSTR genotypes the allele sequence. HipSTR was compared to both GangSTR and ExpansionHunter to genotyping common STR genome wide. Both GangSTR and ExpansionHunter can also identify large STR expansions and were compared against EHdn, STRling and STRetch to assess their ability to call rarer large STR expansions.

These tools have been partially compared in literature by the authors but there is still lack of comprehensive comparison of the STR calling tools. Two additional tools that call STR expansions but are excluded from this study are exSTRa and Tredparse. exSTRa does not estimate the allele length and Tredparse cannot estimate repeat lengths longer than sequencing mate-pair length [20, 25].

The genome data analysed here includes those from the Genome in a Bottle consortium and the 1000 Genomes project, which are collections of publicly-available samples. Our aim in this study is to benchmark different STR-calling software tools primarily using these reference samples. In particular, the Genome in a Bottle consortium provides key reference standard samples analysed by many different methods, so as new technical approaches become available we encourage others to use these same reference samples to facilitate fair comparison of software tools between studies [26].

## Methods

### Ethics statement

All sequencing data used in this study are from previous studies, from fully consented individuals. Sample collections are from the 1000 Genomes project (https://www.internationalgenome.org/sample_collection_principles/), Personal Genome Project Canada (https://personalgenomes.ca/), or NIGMS Human Genetic Cell Repository at the Coriell Institute for Medical Research.

### Samples and datasets

An overview of the approaches used in this paper is shown in Fig 1, and the different features of the software tools are shown in Table 1.

The six tools were benchmarked by analysing three different datasets. Firstly, ten gold-standard high coverage short-read human genome sequences from the Genome in a Bottle consortium (https://github.com/genome-in-a-bottle/giab_data_indexes) consisting of two child-parent trios. Secondly, four child-parent trios randomly selected from the 1000 Genomes project (https://www).internationalgenome.org/data)) and thirdly, 116 samples with known trinucleotide repeat mutations obtained from the Coriell Institute (EGA accession number EGAD00001003562). The accession numbers for samples obtained from the Genome in a Bottle (GIAB) and the 1000 Genomes Project are listed in Table 2.

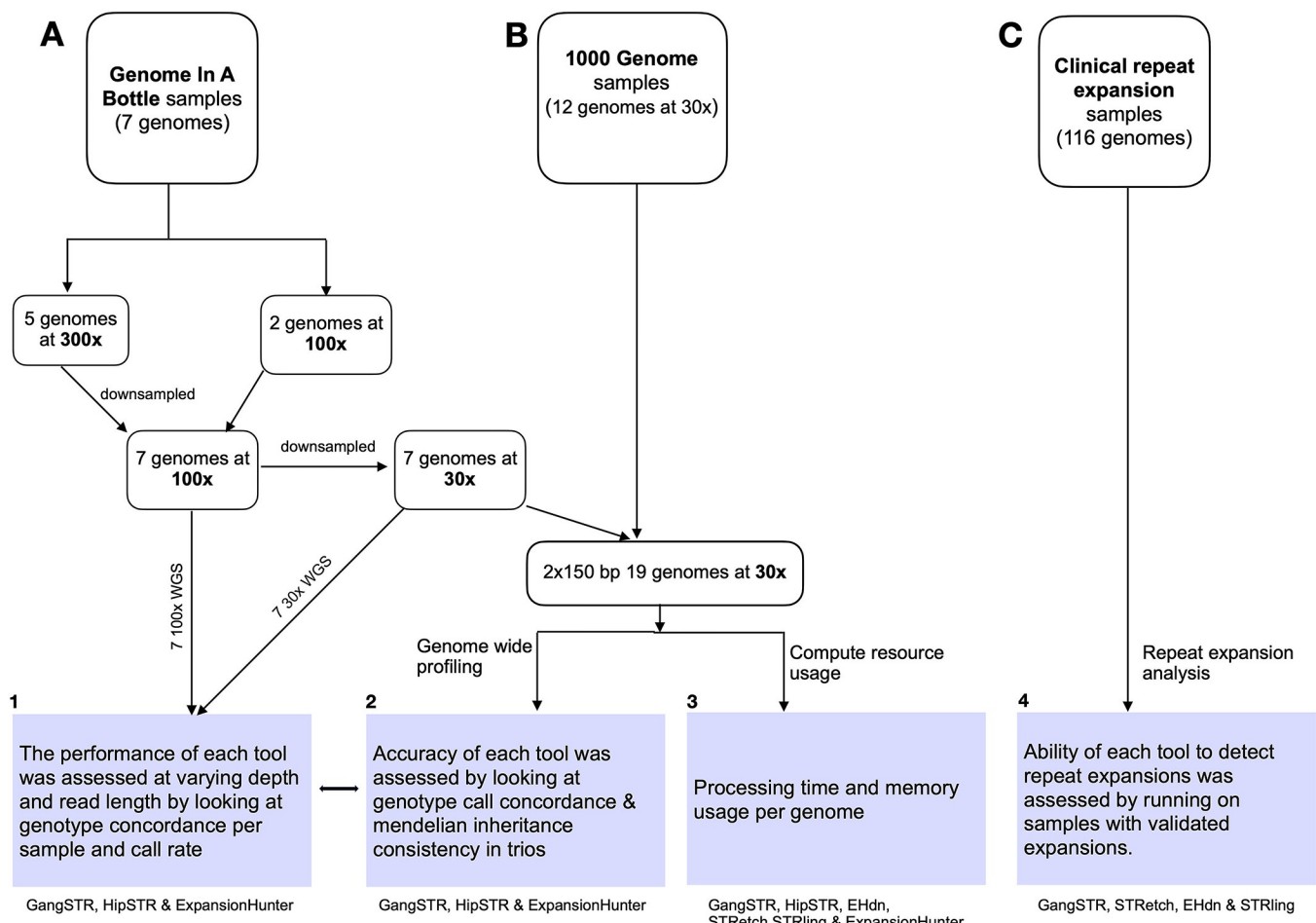

**Fig 1. Flowchart of analyses and samples used in this study.** a) Genome in a bottle (GIAB) samples. b) 1000 Genomes project samples. c) Known clinical samples available from Coriell.

The 116 whole genomes including previously validated disease-causing STR expansions obtained [24] have been validated with one the following repeat expansions: Fragile X Syndrome (FMR1), Huntington disease (HTT), Friedreich's ataxia (FXN), Myotonic Dystrophy

**Table 1. Software for genotyping tandem repeats compared.**

| Software | Repeat unit size range (bp) | STR catalogue required? | Genomewide? | Approach | Estimates repeat length and/or sequence | Reference |
|---|---|---|---|---|---|---|
| HipSTR v.0.6.2 | 1–6 | Yes | Yes; only < read length | Individual calling | Both | Willems et al., 2017 |
| GangSTR v.2.5.4 | 1–20; 20+ | Yes | Yes | Individual calling | Only length | Mousavi et al., 2019 |
| ExpansionHunter v.5.0.0 | 1–6; 6+ | Yes | Yes | Individual calling | Only length | Dolzhenko et., 2017 |
| STRetch | 1–6 | Yes | yes, only expansions | Case/control—outlier | Only length | Dashnow et al., 2018 |
| STRling | 1–6 | No | yes, only expansions | Individual calling & Case/control—outlier | Both | Dashnow et al., 2022 |
| ExpansionHunter Denovo | 2–20; 20+ | No | yes, only expansions | Case/control -outlier | Both | Dolzhenko et al., 2020 |

**Table 2. Samples used in this study.**

| Sample ID | Population | Trio | Collection | Average fold sequence coverage | Paired-end read length (bp) | Accession number |
|-----------|-----------|------|-----------|-------------------------------|----------------------------|-----------------|
| HG002 | Ashkanazi | Son | GIAB | 30, 100, 300 | 150, 250 | NA24385 |
| HG003 | Ashkanazi | Father | GIAB | 30, 100, 300 | 150, 250 | NA24149 |
| HG004 | Ashkanazi | Mother | GIAB | 30, 100, 300 | 150, 250 | NA24143 |
| HG005 | Chinese | Son | GIAB | 30, 100, 300 | 250 | NA24631 |
| HG006 | Chinese | Father | GIAB | 30, 100 | 150 | NA24694 |
| HG007 | Chinese | Mother | GIAB | 30, 100 | 150 | NA24691 |
| NA12878 | CEU | - | GIAB | 30, 100, 300 | 150 | NA12878 |
| HG00403 | CHS | Father | 1000G | 30 | 150 | HG00403 |
| HG00404 | CHS | Mother | 1000G | 30 | 150 | HG00404 |
| HG00405 | CHS | Son | 1000G | 30 | 150 | HG00405 |
| NA18485 | YRI | Son | 1000G | 30 | 150 | NA18485 |
| NA18487 | YRI | Father | 1000G | 30 | 150 | NA18487 |
| NA18489 | YRI | Mother | 1000G | 30 | 150 | NA18489 |
| HG01500 | IBS | Father | 1000G | 30 | 150 | HG01500 |
| HG01501 | IBS | Mother | 1000G | 30 | 150 | HG01501 |
| HG01502 | IBS | Son | 1000G | 30 | 150 | HG01502 |
| NA06984 | CEU | Father | 1000G | 30 | 150 | NA06984 |
| NA06989 | CEU | Mother | 1000G | 30 | 150 | NA06989 |
| NA12329 | CEU | Daughter | 1000G | 30 | 150 | NA12329 |

GIAB = Genome in a Bottle 1000G = 1000 Genomes Project

(DM1), Spinocerebellar Ataxia 1/3 (ATXN1/3), Spinal and Bulbar Muscular Atrophy (SBMA). The samples had been sequenced at 2x150 bp reads on Illumina HiSeqX and repeat expansions previously detected using ExpansionHunter and standard PCR techniques [24]. All samples had been Illumina sequenced using PCR-free methods. The sequencing methods for these data sets have been described in [24, 26, 27].

## Bam file preparation

The seven GIAB bam files were down-sampled from their original coverage of either 300x or 100x to a final coverage of ~30x. In summary, a total of 5/7 GIAB samples had an initial genome coverage of ~300x and 2/7 had a genome coverage of ~100x. The 300x genomes were down-sampled to 100x and all samples were down-sampled further to ~30x genome coverage using Picard v2.6.0 software [28]. The final 30x coverage was chosen to assess the applicability of STR genotyping tools on large cohorts of genomes sequenced at 30x genome coverage, which is a standard coverage adopted by the 1000 Genomes project, for example. The bam files were refined to remove sequence duplicates and the quality of the bam files assessed using Qualimap v2.2.1 [29]. This data set was collated with twelve 30x genomes from the 1000G Project (1 KGP). Overall, we generated 3 datasets: 300x (n = 5), 100x (n = 7) and 30x (n = 19) genomes, including 6 child-parent trios, with three individuals sequenced using both 150bp and 250bp paired-end reads (Table 2). These sequences had been aligned to the GRCh38 human genome assembly [26]. All the bam files were sorted and indexed using samtools v1.9 [30] before calling for STR genotypes.

## Computing memory and run time evaluation

To evaluate compute resource usage, we recorded the time and memory taken to process each bam file from five samples Illumina short-read sequenced at 30x coverage, using a single core

of an Intel Xeon Skylake CPU running at 2.6GHz clock speed with 80Gb RAM available. GangSTR, HipSTR, STRetch and ExpansionHunter were run on a custom STR catalogue containing 811899 STRs of 2–6 bp repeat units. This catalogue was built from GangSTR's catalogue (hg38_v13.bed) available at https://github.com/gymreklab/GangSTR, limited to 2–6 bp repeat units. Because ExpansionHunter is sensitive to the presence of unambiguous 'NNs' in a reference genome around the STR, some of the loci were dropped leaving a total of 790661 loci. This custom catalogue used is available at: https://doi.org/10.25392/leicester.data.22041020. GangSTR, HipSTR, and ExpansionHunter were run at default parameters using this custom catalogue. STRetch was run using whole genome sequencing pipeline starting from mapped bam files [22]. STRling and ExpansionHunter Denovo (EHdn) do not require a reference catalogue. STRling was run using single sample pipeline [23]. Both EHdn and STRetch require a control set of samples. To explore STR expansion profiles in each sample, for EHdn, the sample to be examined was treated as a case sample and the rest of the samples as controls and performed outlier analysis. For STRetch, we build a control set from the remaining subset of the genomes analysed in this study using STRetch pipeline [22]

## Comparison of software for genotyping STRs

HipSTR, GangSTR and ExpansionHunter performance was compared genome-wide by assessing: (a) the proportion of STRs genotyped (b) accuracy of the calls made by analysing Mendelian inheritance patterns in 6 child-parent trios and sample call-concordance compared across varying sequence depths and sequence read lengths. First, each tool was run using its own STR catalogue published and tested on the respective tool. These catalogues are of different sizes, with GangSTR listing 832,380 STR loci, HipSTR listing 1,638,945 loci and ExpansionHunter listing 174,293 loci. These catalogues are available at: https://github.com/HipSTR-Tool/HipSTR, https://github.com/gymreklab/GangSTR and https://github.com/Illumina/RepeatCatalogs. HipSTR and ExpansionHunter catalogues consist of STRs of 2–6 bp units while GangSTR consist of 2-20bp repeat units. GangSTR was run using default parameters. HipSTR was run using some non-default parameters: —min-reads 10 and—def-stutter-model. The parameters were used to set the minimum total reads required to genotype a locus from 100 and allow a default stutter model, recommended for running few samples [19]. ExpansionHunter was run at default parameters, see codes used in data availability section. For comparisons between the three software tools, they were all run using the custom catalogue of 790661 STRs described above. The raw variant calling files (VCF) were filtered using dumpSTR tool to remove calls with read coverage below 10 and those with abnormally high coverage by setting the max call depth for all tools to 1000 [31]. Additional parameters were used for GangSTR:—filter- spanbound-only and—filter-badCI, to filter out calls where only spanning or bounding reads were found, or calls with the maximum likelihood genotype estimates outside of the 95% bootstrap confidence interval as recommended by the GangSTR authors [25]. The performance of the three methods was further assed by comparing the genotype calls to CE data [32] across 13 forensic STRs, known as core Combined DNA Index System (CODIS). This panel was further extended to include 9 additional forensic STRs and assessed the accuracy of the 3 software by assessing the mendelian inheritance patterns in the 5 parent-child trios.

## Comparison of software for detecting and genotyping STR expansions

To assess the sensitivity of GangSTR, EHdn, STRling and STRetch at detecting STR expansions, the four tools were run on samples with known clinical STR expansions that had been analysed by ExpansionHunter in a previous publication [24]. EHdn was run in a case- control mode as opposed to outlier analysis [21]. For each clinical STR expansion locus, the samples

with those STR expansions were treated as cases, and samples with different STR expansions or normal allele sizes used as controls. For STRetch, samples were used as a control to each other, and the depth normalised read counts compared to identify significantly expanded loci in each sample relative to the rest. Both STRetch and GangSTR were run using STR catalogues containing disease loci (https://stripy.org/expansionhunter-catalog-creator). GangSTR was run with the same parameters described above. However, for Fragile X Syndrome and Spinal and Bulbar Muscular Atrophy, GangSTR was run by specifying the ploidy of the X chromosome for male and female. STRling was run using joint calling pipeline [23]. Our results were compared to the previously reported validated calls for these samples [24].

## Results

### Computing resource usage

For large scale studies involving thousands of samples, the relative processing time of different software tools can become an important aspect of software choice. We first compared the software tools that use a STR catalogue (Table 3). GangSTR, HipSTR, ExpansionHunter and STRetch were run on the same STR catalogue containing 790661 loci. GangSTR took an average of 10 hours per diploid genome across five samples, STRetch took about 7 hours and HipSTR took about 3 hours. ExpansionHunter default mode (seeking mode) needed more than 7 days and was not run to completion. For the software tools that do not require a software catalogue, EHdn and STRling took around 30 minutes per diploid genome.

STRetch and ExpansionHunter are the only software tools tested with a multithreading option. For STRetch increasing the core number from 1 to 28 reduces the time from 7 hours to about 1.5 hours per diploid genome. Using ExpansionHunter's streaming mode (that is recommended for large genomic catalogs) with at least 16 threads noticeably improved its performance, reducing its run time to an average of 1.5 hours per diploid genome (Table 3). ExpansionHunter was notable for its high memory usage–typically around 70Gb in contrast to the other 5 software tools using either ~17 Gb or less than 1GB (Table 3).

### Number of common STR loci genotyped and genotype call concordance of common STRs

We compared GangSTR, ExpansionHunter and HipSTR, which are the three software tools that aim to genotype all STRs across the genome. They all rely on a catalogue of STR loci provided by the user, and the number of the loci in that catalogue provide the upper limit on the number of genotyped STRs. Therefore, our results are presented as a percentage of STRs called using that particular catalogue. We first used the different catalogues provided with each

**Table 3. Comparison of processing times and memory usage of STR-calling software.**

| Sample | CPU Time (hours:minutes:seconds) | | | | | | | RAM in Gb | | | | | | |
|---|---|---|---|---|---|---|---|---|---|---|---|---|---|---|
| | GangSTR | HipSTR | STRetch | EH (default mode) | EH (streaming mode) | EHdn | STRling | GangSTR | HipSTR | STRetch | EH (default mode) | EH (streaming mode) | EHdn | STRling |
| NA18485 | 10:33:01 | 3:58:53 | 06:39:38 | > 168 h | 01:33:22 | 00:30:50 | 00:24:33 | 0.04 | 0.50 | 17.63 | n/a | 69.80 | 0.57 | 1.16 |
| NA18487 | 10:27:03 | 3:00:32 | 07:10:47 | > 168 h | 01:28:28 | 00:30:09 | 00:24:03 | 0.03 | 0.49 | 17.60 | n/a | 72.56 | 0.48 | 0.87 |
| NA06984 | 10:13:42 | 2:47:50 | 06:21:20 | > 168 h | 01:23:40 | 00:30:33 | 00:24:19 | 0.04 | 0.40 | 17.42 | n/a | 70.11 | 0.58 | 1.16 |
| NA06989 | 8:41:14 | 2:34:05 | 05:57:05 | > 168 h | 01:17:33 | 00:27:24 | 00:22:11 | 0.04 | 0.42 | 17.51 | n/a | 69.82 | 0.57 | 0.82 |
| NA12329 | 10:24:04 | 2:50:49 | 07:15:59 | > 168 h | 01:19:26 | 00:29:49 | 00:23:42 | 0.05 | 0.34 | 17.40 | n/a | 72.57 | 0.51 | 0.76 |
| Average | 10:03:48 | 03:02:25 | 06:40:57 | n/a | 01:24:29 | 00:29:45 | 00:23:45 | 0.04 | 0.43 | 17.51 | n/a | 70.97 | 0.54 | 0.96 |

software tool, and found several patterns common to both GangSTR, ExpansionHunter and HipSTR (S1 Table). Firstly, all software tools call a high proportion of the STRs in their respective catalogues across all samples, with GangSTR and ExpansionHunter calling a higher proportion than HipSTR, although the different sizes of their catalogues must be taken into account. Secondly, more calls are made from 100x coverage data than 30x. The increase is small across most samples but is large (over 10 percentage points) for HG005 for HipSTR and GangSTR, where both tools call fewer than 90% of STRs in the 30x coverage data. Taken together, 30x is not only sufficient but appears optimal for STR calling, with little benefit in increasing sequencing depth. For read length, 150bp paired-end is optimal and increased read length seems slightly detrimental, at least for HipSTR, which made fewer calls across the four samples that were sequenced at 2x250bp.

To ensure a fair comparison across tools, we repeated our analysis using the same STR catalogue containing 790661 loci (Table 4). For all software tools and samples, 90% of STRs in this catalogue were called, with most calling >99%. We compared the actual genotype calls made on the STRs called by three tools by plotting each allele call for each on a scatterplot, normalised against the allele in the reference genome (Fig 2). This shows that over 85% of the calls were identical between the three methods, and of those calls, at least 89.0% were identical with the reference genome. There was no strong bias in under- or over-calling repeat number made by HipSTR, ExpansionHunter or GangSTR. There was high concordance of the genotype calls made at 2x150bp compared to 2x250bp for each tool (S1 Table).

## Accuracy of common STR calls assessed using Mendelian inheritance

Accuracy of the genotypes made by GangSTR, ExpansionHunter and HipSTR can be measured using the expectation of Mendelian inheritance of alleles, whereby we would expect to see one allele of a child's genotype in their mother and one allele in their father. Any allele in a child not observed in one of the parents is due either to a genotyping error in the child or the parent, or a de novo mutation in the child. Assuming a rate of de novo mutations at STR loci as $10^{-4}$ per generation [33], the contribution to observed inheritance errors of de novo mutation is minimal and therefore differences in inconsistencies can be attributed to differences in STR genotyping accuracy.

Analysis of Mendelian inconsistencies using bcftools software across the five mother-father-child trios with 150bp sequence reads shows very little difference in genotyping accuracy across the three methods. GangSTR and HipSTR showed the highest genotyping accuracy (Table 5). Comparison between the HG002,HG003,HG004 trio sequenced at 100x coverage and 30x coverage shows that increased sequencing coverage does not improve genotyping accuracy.

In order to investigate the possible reasons for erroneous genotype calls, we stratified the genotypes that showed Mendelian inconsistencies by repeat unit size normalised against the repeat-unit counts in the reference list (Fig 3). It is clear that for the three tools; ExpansionHunter, HipSTR and GangSTR, there is overrepresentation of 2 bp repeat unit STRs in the incorrect genotype calls. Therefore, for these methods 2 bp repeat STR genotypes have the highest error rate, over twice the error rate of other STRs (S1 Fig).

## Genotyping of known forensic STR loci

Thirteen STRs, known as core Combined DNA Index System (CODIS) STRs, are long STRs that show extensive variation between individuals, and are used across many different forensic STR panels for identification of individuals from genomic DNA [34, 35]. Analysis of the performance of GangSTR, ExpansionHunter and HipSTR in calling these STRs in comparison to

**Table 4. Percentage of loci called in a shared STR catalogue by GangSTR and HipSTR and ExpansionHunter.**

| Sample | Read length | GangSTR | | | | | HipSTR | | | | | ExpansionHunter | | | | |
|---|---|---|---|---|---|---|---|---|---|---|---|---|---|---|---|---|
| | | Called at 30x (%) | Called at 100x (%) | Calls in both (%) | Identical by allele length (%) | Identical by allele sequence (%) | Called at 30x (%) | Called at 100x (%) | Calls in both (%) | Identical by allele length (%) | Identical by allele sequence (%) | Called at 30x (%) | Called at 100x (%) | Calls in both (%) | Identical by allele length (%) | Identical by allele sequence (%) |
| HG002 | 2x150 | 99.7 | 99.7 | 99.6 | 99.4 | NA | 96.2 | 96.0 | 95.5 | 99.8 | 99.9 | 99.8 | 99.9 | 99.8 | 98.9 | NA |
| HG003 | 2x150 | 99.7 | 99.7 | 99.6 | 99.4 | NA | 96.1 | 96.0 | 95.5 | 99.8 | 99.9 | 99.8 | 99.9 | 99.8 | 99.8 | NA |
| HG004 | 2x150 | 99.2 | 99.2 | 99.1 | 99.4 | NA | 95.8 | 95.5 | 95.2 | 99.8 | 99.9 | 99.3 | 99.5 | 99.3 | 98.9 | NA |
| HG005 | 2x250 | 98.5 | 99.6 | 98.4 | 99.6 | NA | 93.9 | 94.3 | 92.9 | 99.7 | 99.1 | 99.8 | 99.9 | 99.8 | 98.4 | NA |
| HG006 | 2x150 | 99.4 | 99.5 | 99.3 | 99.3 | NA | 96.0 | 95.9 | 95.0 | 99.8 | 99.9 | 99.7 | 99.9 | 99.7 | 98.7 | NA |
| HG007 | 2x150 | 99.0 | 99.1 | 98.9 | 99.3 | NA | 95.7 | 95.4 | 95.0 | 99.8 | 99.8 | 99.3 | 99.5 | 99.3 | 98.7 | NA |
| NA12878 | 2x150 | 99.1 | 99.1 | 98.9 | 99.3 | NA | 95.7 | 95.4 | 95.0 | 99.8 | 99.8 | 99.3 | 99.5 | 99.3 | 98.8 | NA |
| HG002 | 2x250 | 96.1 | NA | NA | NA | NA | 94.1 | NA | NA | NA | NA | 99.3 | NA | NA | NA | NA |
| HG003 | 2x250 | 95.8 | NA | NA | NA | NA | 93.9 | NA | NA | NA | NA | 99.8 | NA | NA | NA | NA |
| HG004 | 2x250 | 96.8 | NA | NA | NA | NA | 93.9 | NA | NA | NA | NA | 99.4 | NA | NA | NA | NA |

NA: not analysed

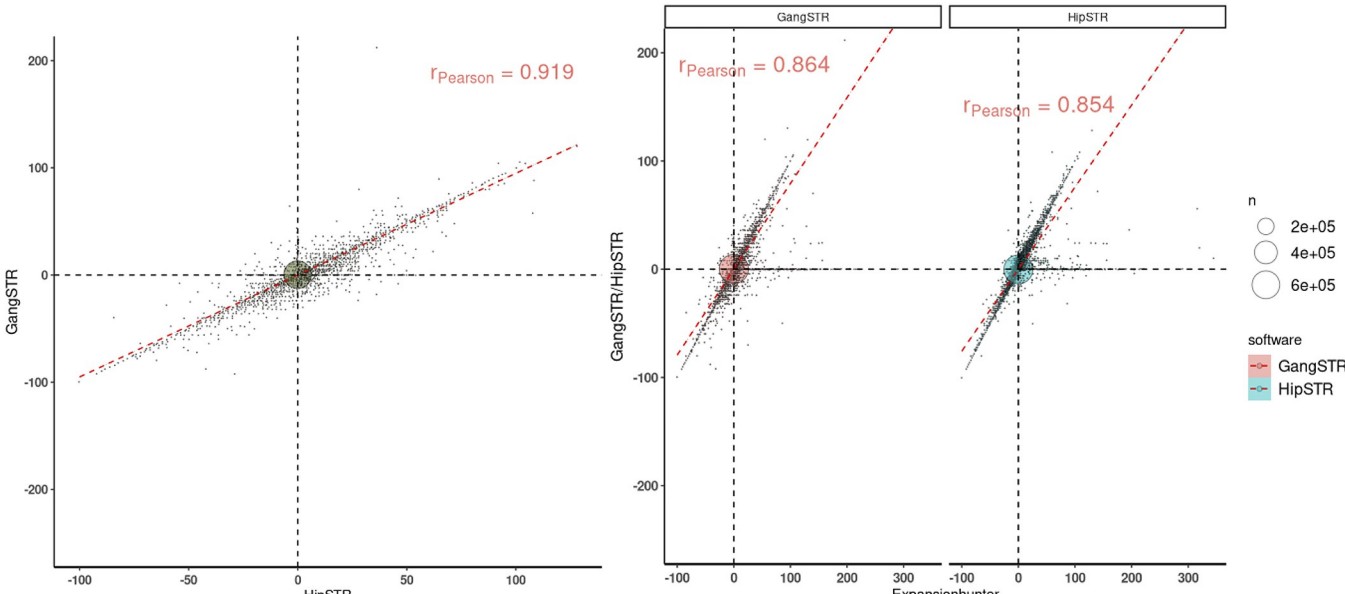

**Fig 2. Comparison of STR calls from ExpansionHunter, HipSTR and GangSTR.** A). Comparison of the HG002 genotypes of the STRs called by both GangSTR and HipSTR. The x -axis represents the call made by GangSTR in comparison to the reference sequence (0 = same as reference sequence, -100 represents 100 fewer repeat units than the reference sequence. The y-axis represents the call made by HipSTR. B) Comparison of the genotypes of the STRs called by either GangSTR or HipSTR compared to ExpansionHunter (x-axis). The dotted red lines shows correlation between the calls compared.

capillary electrophoresis (CE) data is useful both for practical forensic analysis, but also as a measure of the error rate of the software, if we accept that capillary electrophoresis can be regarded as a gold standard. Matched capillary electrophoresis data for the 13 core STRs, generated using forensic-standard Promega Powerplex Fusion 24 assay, has been published [32], and our calls for NA12878 were compared against these data (Table 6). ExpansionHunter calls matched CE data at 10/13 loci, called one allele incorrectly at the THO1 locus and two alleles at both FGA and D21S11 loci. GangSTR genotypes matched CE data for 11/13 loci, with one locus (D21S11) not called and one (TH01) called incorrectly. HipSTR showed the same results as GangSTR, except that, in addition, it incorrectly called one allele at D13S317.The three tools

**Table 5. Mendelian inheritance of STR alleles.**

| Parent-offspring Trio | Coverage | GangSTR Mendelian consistent (%) | HipSTR Mendelian consistent (%) | ExpansionHunter Mendelian consistent (%) |
|---|---|---|---|---|
| HG002,HG003, HG004 | 100x | 99.5 | 99.6 | 99.2 |
| HG002,HG003, HG004 | 30x | 99.7 | 99.5 | 98.7 |
| NA18485, NA18489,NA18487 | 30x | 99.6 | 99.4 | 98.5 |
| NA06984, NA06989,NA12329 | 30x | 99.8 | 99.9 | 98.9 |
| HG00403, HG00404, HG00405 | 30x | 99.9 | 99.9 | 99.0 |
| HG01500, HG01501, HG01502 | 30x | 99.7 | 99.6 | 98.7 |

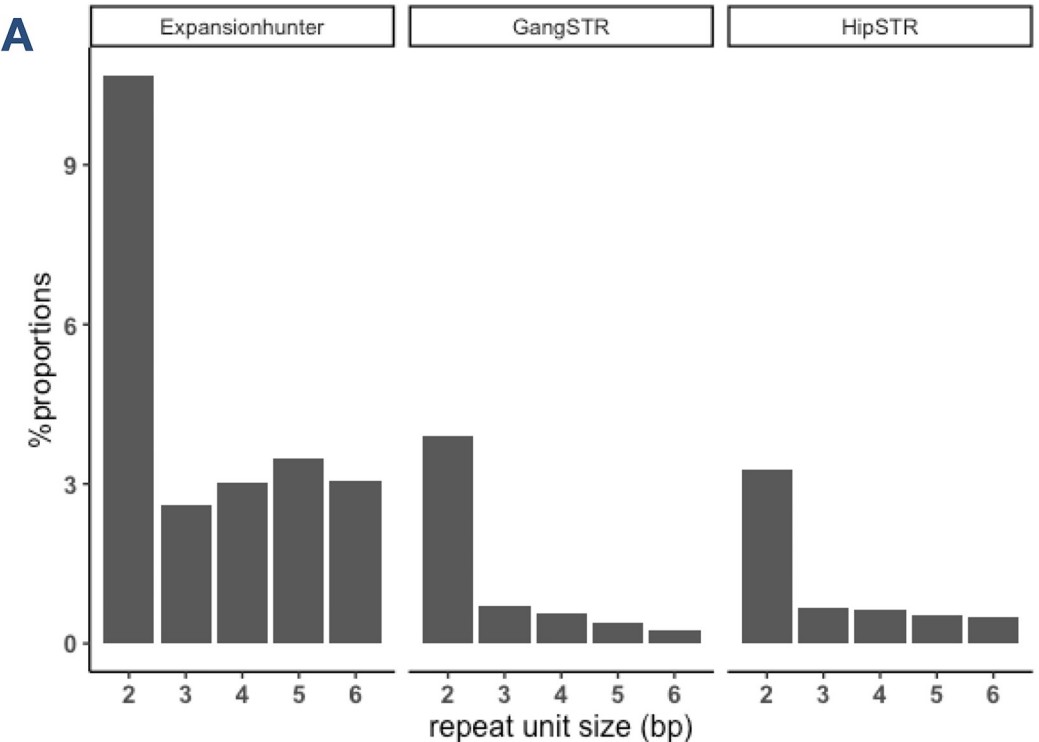

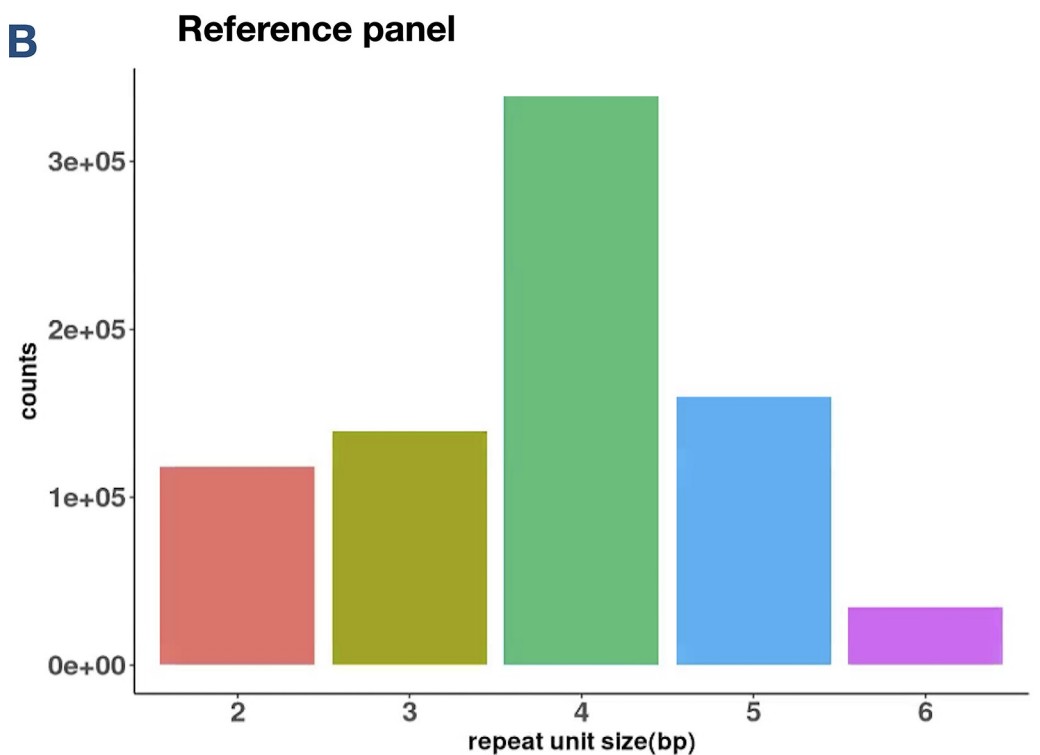

**Fig 3. Repeat unit size distribution of Mendelian inconsistencies.** a) ExpansionHunter, GangSTR and HipSTR calls that are Mendelian inconsistent as a proportion of motif counts in the catalogue, b) all STRs in the catalogue used by the three software tools.

**Table 6. Comparison of STR genotype calls at core forensic loci for NA12878.**

| Locus | Chrom | Start | End | Motif | GangSTR 100x | GangSTR 30x | HipSTR 100x | HipSTR 30x | ExpansionHunter 100x | ExpansionHunter 30x | CE |
|---|---|---|---|---|---|---|---|---|---|---|---|
| CSF1PO | 5 | 150076324 | 150076375 | ATCT | 10,11 | 10,11 | 10,11 | 10,11 | 10,11 | 10,11 | 10,11 |
| D5S818 | 5 | 123775556 | 123775599 | ATCT | 12,12 | 12,12 | 12,12 | 12,12 | 12,12 | 12,12 | 12,12 |
| D7S820 | 7 | 84160226 | 84160277 | TATC | 8,10 | 8,10 | 8,10 | 8,10 | 8,10 | 8,10 | 8,10 |
| D13S317 | 13 | 82148025 | 82148068 | TATC | 11,12 | 11,12 | 11.3,12.3 | 11.3,12.3 | 11,12 | 11,12 | 11,12 |
| D16S539 | 16 | 86352702 | 86352745 | GATA | 10,11 | 10,11 | 10,11 | 10,11 | 10,12 | 10,11 | 10,11 |
| D21S11 | 21 | 19181973 | 19182099 | TCTA | - | - | - | - | 18,34 | 34,34 | 30,30 |
| TH01 | 11 | 2171088 | 2171115 | AATG | 7,8 | 7,7 | 7,9.8 | 7,9.8 | 7,10 | 7,10 | 7,9.3 |
| TPOX | 2 | 1489653 | 1489684 | AATG | 8,8 | 8,8 | 8,8 | 8,8 | 8,8 | 8,8 | 8,8 |
| vWA | 12 | 5983977 | 5984044 | AGAT | 15,17 | 15,15 | 15,17 | 15,17 | 15,17 | 15,17 | 15,17 |
| D3S1358 | 3 | 45540739 | 45540802 | TCTA | 16,17 | 16,17 | 16,17 | 16,17 | 16,17 | 16,17 | 16,17 |
| D8S1179 | 8 | 124894865 | 124894916 | TATC | 12,12 | 12,12 | 12,12 | 12,12 | 12,13 | 12,13 | 12,12 |
| D18S51 | 18 | 63281667 | 63281738 | AGAA | 16,17 | 16,17 | 16,17 | 16,17 | 16,17 | 16,17 | 16,17 |
| FGA | 4 | 154587736 | 154587823 | GGAA | 22,24 | 22,24 | 22,24 | 22,24 | 23,25 | 23,25 | 22,24 |

N/B: F = failed QC; CE = capillary electrophoresis genotype

were also consistent across 19/22 forensic STRs analysed for Mendelian inheritance errors across the six trios.

## Detection of STR expansions at known clinical loci

To assess the reliability in calling expanded STRs in a clinical situation, we compared STRetch, EHdn and GangSTR performance in calling expanded STRs in samples with known STR expansions (Table 7). All four methods showed high sensitivity, with EHdn (95% sensitivity), GangSTR (89% sensitivity) and STRling (94% sensitivity) outperforming STRetch (68% sensitivity). Although STRetch, STRling and GangSTR were able to flag repeat expansions in most of the genes analysed, the three tools underestimated the repeat lengths in comparison to Southern blot results at *DMPK, FMR1* and *FXN* loci, which may be due to either mosaicism of

**Table 7. Detection of STR expansions at known clinical loci.**

| Disease | Gene | Total analysed | Identified using EHdn | Identified using STRetch | Identified using GangSTR | Identified using STRling |
|---|---|---|---|---|---|---|
| Spinal-bulbar muscular atrophy | AR | 1 | 0 | 0 | 0 | 0 |
| Myotonic dystrophy type 1 | DMPK | 16 | 16 | 16 | 16 | 16 |
| Fragile X | FMR1 | 34 | 34 | 19 | 32 | 33 |
| Friedreich's ataxia | FXN | 25 | 25 | 11 | 19 | 24 |
| Huntington's disease | HTT | 13 | 13 | 13 | 13 | 12 |
| Dentatorubral-pallidoluysian atrophy | ATN1 | 2 | 2 | 2 | 1 | 2 |
| Spinocerebellar ataxia type 1 | ATXN1 | 3 | 0 | 3 | 3 | 1 |
| Spinocerebellar ataxia type 3 | ATXN3 | 1 | 0 | 1 | 1 | 1 |
| Total | | 95 | 90 (95%) | 65 (68%) | 85 (89%) | 89 (94%) |
| Unaffected | | 21 | 0 | 0 | 1 | 2 |

the STR expansion or to GC-bias of the Illumina sequencing approach [24]. These loci were characterised by longer and or double expansions in both alleles ranging between 50 to 1000 repeat units in the samples that were screened. Both STRetch and STRling performed poorly at the *FMR1* locus (Table 7). Of the samples identified as expanded (p< = 0.05; adjusted for multiple testing) at the *FMR1* locus by STRetch, all had reported repeat lengths below the expected *FMR1* premutation ranges [5]. For STRling, half the samples identified had reported repeat lengths below the expected *FMR1* premutation ranges. To assess specificity, 21 samples with known non-pathogenic allele length at *FMR1* were analysed, with STRling called two of these as expanded and GangSTR identifying one sample with an expanded allele length. This is a small dataset to robustly test false positive rate, but suggests overall high specificity for the three software tools used.

## Discussion

Genomewide analysis of STRs using short read sequencing has lagged behind studies of other variations, including single nucleotide variation and, to a certain extent, structural variation. It is likely that STRs underly some undiscovered genetic associations with complex disease [5]. Several software tools have been developed with the aim of accurately calling STR genotypes genome-wide, but with different ultimate aims. Some have been developed to detect and genotype large repeat expansions that are outliers from the population distribution of allele lengths for that particular STR. Development of these was motivated by well- established repeat expansions causing a variety of Mendelian diseases, and more recent discoveries showing that large repeat expansions can underlie a large amount of complex disease, such as ALS [36, 37]. Other software tools aim to genotype STRs genome-wide irrespective of their alleles, using a catalogue of known STRs.

We selected the most recent software tools most appropriate to our needs in detecting both STR expansions and genome-wide STR genotypes in a population-based sample. A previous study describes detection of six disease-causing STR expansions from whole exome sequences using several STR-calling software tools, including GangSTR and STRetch but not EHdn and STRling [38]. ExpansionHunter [39], STRetch and exSTRa [40] were used to detect clinically-related STR expansions in those 6 genes with high specificity. Other reports include those focused on forensic STR detection [41], or are focused on genotyping a particular, or small number, of STR loci [42].

Our aim was to assess the feasibility of applying these software tools, both in terms of quality of the resulting data and practicality of using the software, for large cohorts of genomes. We assessed this in a variety of ways. Processor and memory usage, in conjunction with processing time, was measured so that we could assess the feasibility of scaling up analysis to thousands of genomes given current computing resources. For the software using a catalogue to genotype known STRs (GangSTR, HipSTR and Expansionhunter), we examined the effect of increasing sequencing depth on the proportion of STRs genotyped, and accuracy. We then examined the concordance between GangSTR, ExpansionHunter and HipSTR using common STR catalogues, and the relative number of Mendelian inconsistencies in trios to assess STR genotyping accuracy. We found sequencing depth to have no effect on the number or quality of STR genotypes called by HipSTR, but higher sequencing depth had a modest positive effect on GangSTR STR genotype calling. Quality of STR calls made by Expansionhunter, GangSTR and HipSTR was very similar but slightly higher between GangSTR and HipSTR when given the same STR catalogue. Both GangSTR and HipSTR used less memory than Expansionhunter, but GangSTR took about 3x more CPU time. Both GangSTR and HipSTR performed very well, but not perfectly, in genotyping STR loci used for forensics.

The software tools STRetch, STRling, ExpansionHunter, EHdn and GangSTR can all call expanded repeats, with only EHdn and STRling not requiring a predefined catalogue of STR loci. Because expanded repeats are expected to be rare, and underlie several clinical conditions, we used a previous dataset to test the ability of STRetch, STRling, EHdn and GangSTR to genotype known STR expansions. EHdn, STRling and GangSTR show higher sensitivity than STRetch, at least under the conditions tested. EHdn and STRling used the least resource, analysing a genome in about half an hour using about half a Gb of memory under our conditions.

For case-control studies where thousands of genomes have been sequenced at ~30x coverage for both large expansions and known STRs genomewide, we have decided to use GangSTR for known STRs and EHdn for large expansions. Although STRling used similar time and memory as EHdn when run in a single sample mode, its resource usage can increase linearly in joint calling mode which is appropriate for case-control outlier analysis [23]. EHdn and STRling are the only STR genotypers that can genotype expansions without a prior defined catalogue, broadening their scope to identify previously unknown expansions. GangSTR and HipSTR are similar, with the increased computing resources needed by GangSTR offset in our view by its ability to detect larger STR expansions longer than the read length, to support and extend EHdn calls. ExpansionHunter default mode was unusable at genotyping a catalog of 790661 STR loci as it needed more than 7 days but efficient in streaming mode albeit higher memory requirements about 70 Gb per genome using 16 cores.

We note that, although the clinical repeat expansions and the forensic STRs have been validated by orthogonal data and methods, this is not the case with the other STR genotypes. For STRs that are associated with disease in subsequent analyses, it will be important to validate those particular STRs on a subset of samples using alternative methods, such as capillary electrophoresis. STR genotype calling from high throughput sequencing remains an area of active development, and we hope that further progress can be made in reliably calling STRs from both short read and long read sequencing data.

Our study had some limitations inherent to different tools. First, we did not analyse the 1bp repeat unit motifs. Both GangSTR and EHdn have been optimised to genotype 2-20bp repeat motifs, of which EHdn only genotypes expanded repeats. 1bp repeat motifs present a challenge to many sequencing technologies due to PCR stutter noise and may introduce artefacts in sequence, however, a detailed assessment of accuracy of 1bp repeat motif STR genotyping in PCR-free sequencing is needed. Secondly, apart from EHdn and STRling, the tools require defined STR coordinates built from reference genomes. Therefore, STRs not assembled in the reference genome, including the highly complex regions of the genome that are often hard to sequence and assemble, cannot be genotyped. However, with improvements in PCR free protocols and long read sequencing, combined with new genome assemblies, the ability to genotype these regions will improve.

## Supporting information

**S1 Table. Number of loci called as percentage of total in catalogue for GangSTR and HipSTR and ExpansionHunter, and call concordance as function of sequence depth.** (DOCX)

**S2 Table. Number of loci called as percentage of total in catalogue for GangSTR and HipSTR and ExpansionHunter, and call concordance as function of sequence length.** (DOCX)

**S1 Fig. ExpansionHunter, GangSTR and HipSTR's 2 bp repeat unit calls—Mendelian inconsistent sites stratified by sequence motif.** (TIFF)

## Acknowledgments

The views expressed are those of the authors and not necessarily those of the National Health Service (NHS), the NIHR or the Department of Health. This research used the ALICE High Performance Computing Facility at the University of Leicester.

## Author Contributions

**Conceptualization:** John W. Oketch, Louise V. Wain, Edward J. Hollox.

**Data curation:** John W. Oketch.

**Formal analysis:** John W. Oketch.

**Funding acquisition:** John W. Oketch, Louise V. Wain, Edward J. Hollox.

**Investigation:** John W. Oketch, Edward J. Hollox.

**Methodology:** John W. Oketch.

**Supervision:** Louise V. Wain, Edward J. Hollox.

**Writing – original draft:** John W. Oketch, Edward J. Hollox.

**Writing – review & editing:** John W. Oketch, Louise V. Wain, Edward J. Hollox.

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
