## [Decision Letter · Decision Letter 0]

15 Feb 2024

PONE-D-24-00164A comparison of software for analysis of rare and common short tandem repeat (STR) variation using human genome sequences from clinical and population-based samplesPLOS ONE

Dear Dr. Hollox,

Thank you for submitting your manuscript to PLOS ONE. After careful consideration, we feel that it has merit but does not fully meet PLOS ONE’s publication criteria as it currently stands. Therefore, we invite you to submit a revised version of the manuscript that addresses the points raised during the review process.

We look forward to receiving your revised manuscript.

Kind regards,

Paul Aurelian Gagniuc, PhD

Academic Editor

PLOS ONE

Journal Requirements:

“JWO is funded by a Wellcome Trust PhD studentship as part of the Wellcome Trust Genetic Epidemiology and Public Health Genomics Doctoral Training Programme by grant number 218505/Z/19/Z . LWV holds a GSK/Asthma+Lung UK Chair in Respiratory Research (C17-1). The research was partially supported by the National Institute for Health Research (NIHR) Leicester Biomedical Research Centre

“JWO is funded by a Wellcome Trust PhD studentship as part of the Wellcome Trust Genetic 515 Epidemiology and Public Health Genomics Doctoral Training Programme by grant number 516 218505/Z/19/Z . LWV holds a GSK/Asthma+Lung UK Chair in Respiratory Research (C17-1). The 517 research was partially supported by the National Institute for Health Research (NIHR) Leicester 518 Biomedical Research Centre; the views expressed are those of the author(s) and not necessarily 519 those of the National Health Service (NHS), the NIHR or the Department of Health.”

“JWO is funded by a Wellcome Trust PhD studentship as part of the Wellcome Trust Genetic Epidemiology and Public Health Genomics Doctoral Training Programme by grant number 218505/Z/19/Z . LWV holds a GSK/Asthma+Lung UK Chair in Respiratory Research (C17-1). The research was partially supported by the National Institute for Health Research (NIHR) Leicester Biomedical Research Centre”

4. Please update your submission to use the PLOS LaTeX template. The template and more information on our requirements for LaTeX submissions can be found at http://journals.plos.org/plosone/s/latex.

Reviewers' comments:

Reviewer's Responses to Questions

**Comments to the Author**

1. Is the manuscript technically sound, and do the data support the conclusions?

Reviewer #1: Yes

Reviewer #2: Yes

2. Has the statistical analysis been performed appropriately and rigorously? 

Reviewer #1: Yes

Reviewer #2: Yes

3. Have the authors made all data underlying the findings in their manuscript fully available?

Reviewer #1: Yes

Reviewer #2: Yes

4. Is the manuscript presented in an intelligible fashion and written in standard English?

Reviewer #1: Yes

Reviewer #2: Yes

5. Review Comments to the Author

Reviewer #1: This study is important as it identifies an approach to investigate the role of genomic STR variants in polygenic human diseases. It is also a study that genotypes common STRs with the software used and identifies rarer STR expansions genome-wide. Therefore, it can be published in the journal.

Tables must be written in the appropriate format. There is duplicate text content in the article writing, it was uploaded twice. It should be reviewed as a writing language.

Reviewer #2: The authors present a benchmarking of open-source software for the analysis of repeat expansion from short-read genomic data. The comparison is made considering: computational resources, genotyping accuracy (expansion length and variant calling). Four cases of application at genome wide detection, mendelian inheritance (allele differentiation), rare disease (known pathogenic expansions) and in forensic analysis (variant calling) are described. The experiments performed and the well detailed results are very useful for other scientists working with repeat expansion detection and encourage the use of good practices. The research topic fits the current scenario, as genomic sequencing is becoming cheaper and the availability of genomic data for analysis in different research fields (large population studies, clinical diagnostics, etc.) is increasing. Comparative evaluation of tools for detection of repeat expansions has not been widely published in the past. The comparative evaluation of such variants is of great importance to apply correct bioinformatics approaches in routine laboratory work, due to their genomic variability at the sequence level and their importance in diseases.

The abstract and introduction are well presented addressing the current state of the art of the capabilities of detection tools. The experiments used are adequate and cover all the performance-based questions that the authors aim to address. The experimental design is well conceived control samples are grouped according to the questions. The design of the study considers differences between software that could imply restrictions in the comparison of the results. The number of samples used is sufficient to obtain meaningful results and the reference materials chosen (coriell and 1000genomes) are adequate. For the comparison in each category, performance parameters were well chosen. The tables and figures are understandable and help other scientists to consider which software might be best applied to specific scientific questions. The study conforms to ethical standards. The data are freely available in repositories (web links work) and the methods are available for replication of the results by any other user. Data collection and interpretation is well done.

The presentation of the results is well structured and explained. It supports the conclusions and discussion. Supplementary tables provide all the data produced in the study at sample and variant level and provide sufficient evidence to support the benchmarking goals. The discussion is conducted in the context of the results presented and at the current or research level, limitations are discussed. The statistical analysis and parameters chosen for benchmarking conform to current guidelines for quantitative and qualitative variant detection. Figures and tables are well presented and support the results.

The use cases in Mendelian disorders and forensic medicine focus on current issues in STR detection (i.e. allele-specific genotyping and call accuracy, respectively). The clinical samples FMR1, HTT and FXN are good examples, as they have proven to be tricky in accurate detection. The authors point out the limitations of the study, such as the number of samples was not significant enough (e.g. FMR1) and the type of variation they do not include (e.g. 1 bp).

Overall, this study does not present a groundbreaking advance in the field but presents a comparative assessment for a type of genomic variant that is difficult to determine and for many years overlooked in population and clinical studies. Such benchmarking studies are urgently needed to improve best practice in selecting strategies for genomic analysis, especially in clinical settings. Importantly, the authors also addressed computational resources, an issue that is often underestimated and results in unexpectedly high costs.

Comments for the authors:

The authors state in the Abstract: “Their contribution to common disease is not well-understood, but recent software tools designed to genotype STRs using short read sequencing data are beginning to address this.” Referring to repeats. Software help to address questions related to repeat contribution to common disease, but the software itself does not address contribution. The sentence is misleading and should be changed to improve the meaning within the context of the abstract.

Authors should not refer to Tables (Table 1 is mentioned twice) or detailed results in the introduction. Please delete.

Authors should more clearly specify in the introduction that the manuscript focuses exclusively on short read sequencing data (and not long read sequencing).

Figure 3 resolution is pixelized in the review document. Please make that figures have the required resolution for publication.

Authors assume as a gold standard capillary electrophoresis for the genotyping of known forensic STR loci to benchmark error rates. The accuracy of this technique should be stated (sensitivity specificity) and discussed in case this might have an influence in the cases where calls do not match with the software.

Authors show underestimation of repeat length in clinical cases with very large expansions. Please briefly discuss which are the limitations in the software or short read techniques that cause these underestimations.

6. PLOS authors have the option to publish the peer review history of their article (what does this mean?). If published, this will include your full peer review and any attached files.

Reviewer #1: No

Reviewer #2: No

---

## [Author Response · Author response to Decision Letter 0]

26 Feb 2024

Response to reviewers

Reviewer #1: This study is important as it identifies an approach to investigate the role of genomic STR variants in polygenic human diseases. It is also a study that genotypes common STRs with the software used and identifies rarer STR expansions genome-wide. Therefore, it can be published in the journal.

Tables must be written in the appropriate format. There is duplicate text content in the article writing, it was uploaded twice. It should be reviewed as a writing language.

Thanks to the reviewer for their positive comments. Indeed, the entire article was presented twice in the package to reviewers – this was a mistake. We will follow the publisher’s guidance on table formatting.

Reviewer #2: The authors present a benchmarking of open-source software for the analysis of repeat expansion from short-read genomic data. The comparison is made considering: computational resources, genotyping accuracy (expansion length and variant calling). Four cases of application at genome wide detection, mendelian inheritance (allele differentiation), rare disease (known pathogenic expansions) and in forensic analysis (variant calling) are described. The experiments performed and the well detailed results are very useful for other scientists working with repeat expansion detection and encourage the use of good practices. The research topic fits the current scenario, as genomic sequencing is becoming cheaper and the availability of genomic data for analysis in different research fields (large population studies, clinical diagnostics, etc.) is increasing. Comparative evaluation of tools for detection of repeat expansions has not been widely published in the past. The comparative evaluation of such variants is of great importance to apply correct bioinformatics approaches in routine laboratory work, due to their genomic variability at the sequence level and their importance in diseases.

The abstract and introduction are well presented addressing the current state of the art of the capabilities of detection tools. The experiments used are adequate and cover all the performance-based questions that the authors aim to address. The experimental design is well conceived control samples are grouped according to the questions. The design of the study considers differences between software that could imply restrictions in the comparison of the results. The number of samples used is sufficient to obtain meaningful results and the reference materials chosen (coriell and 1000genomes) are adequate. For the comparison in each category, performance parameters were well chosen. The tables and figures are understandable and help other scientists to consider which software might be best applied to specific scientific questions. The study conforms to ethical standards. The data are freely available in repositories (web links work) and the methods are available for replication of the results by any other user. Data collection and interpretation is well done.

The presentation of the results is well structured and explained. It supports the conclusions and discussion. Supplementary tables provide all the data produced in the study at sample and variant level and provide sufficient evidence to support the benchmarking goals. The discussion is conducted in the context of the results presented and at the current or research level, limitations are discussed. The statistical analysis and parameters chosen for benchmarking conform to current guidelines for quantitative and qualitative variant detection. Figures and tables are well presented and support the results.

The use cases in Mendelian disorders and forensic medicine focus on current issues in STR detection (i.e. allele-specific genotyping and call accuracy, respectively). The clinical samples FMR1, HTT and FXN are good examples, as they have proven to be tricky in accurate detection. The authors point out the limitations of the study, such as the number of samples was not significant enough (e.g. FMR1) and the type of variation they do not include (e.g. 1 bp).

Overall, this study does not present a groundbreaking advance in the field but presents a comparative assessment for a type of genomic variant that is difficult to determine and for many years overlooked in population and clinical studies. Such benchmarking studies are urgently needed to improve best practice in selecting strategies for genomic analysis, especially in clinical settings. Importantly, the authors also addressed computational resources, an issue that is often underestimated and results in unexpectedly high costs.

We thank the reviewer for their careful review and positive comments. Changes are highlighted under “track changes” in the manuscript document.

Comments for the authors:

The authors state in the Abstract: “Their contribution to common disease is not well-understood, but recent software tools designed to genotype STRs using short read sequencing data are beginning to address this.” Referring to repeats. Software help to address questions related to repeat contribution to common disease, but the software itself does not address contribution. The sentence is misleading and should be changed to improve the meaning within the context of the abstract.

We have changed “Their contribution to common disease is not well-understood, but recent software tools designed to genotype STRs using short read sequencing data are beginning to address this.” to “Their contribution to common disease is not well-understood, but recent software tools designed to genotype STRs using short read sequencing data will help address this.”

Authors should not refer to Tables (Table 1 is mentioned twice) or detailed results in the introduction. Please delete.

We have deleted references to table 1 in the introduction, and have cut out two sentences that discuss the results in detail (lines 124-128).

Authors should more clearly specify in the introduction that the manuscript focuses exclusively on short read sequencing data (and not long read sequencing).

We agree, and have made two clarifications in the introduction : line 111, line 118, and one in the abstract (line 35), and twice in the methods (lines 165, 205).

Figure 3 resolution is pixelized in the review document. Please make that figures have the required resolution for publication.

Resolution of figures will be publication quality, following the publisher’s requirements.

Authors assume as a gold standard capillary electrophoresis for the genotyping of known forensic STR loci to benchmark error rates. The accuracy of this technique should be stated (sensitivity specificity) and discussed in case this might have an influence in the cases where calls do not match with the software.

STR calling using capillary electrophoresis will, of course, have an error rate. Sensitivity and specificity will be very dependent on sample origin, quality and assay. However, these data had been generated using a forensic genotyping kit (Promega’s Powerplex Fusion) which will be very robust, and optimised for use on poor-quality forensic samples. Therefore we believe that, when used on laboratory quality samples, we are justified in using these genotypes as an error-free gold standard. We have emphasised this in the manuscript by including “forensic-standard” (line 392).

Authors show underestimation of repeat length in clinical cases with very large expansions. Please briefly discuss which are the limitations in the software or short read techniques that cause these underestimations. 

We echo Dolzhenko et al 2017, in suggesting that GC-bias of Illumina sequencing (leading to a lower coverage than expected of GC-rich repeats), or somatic mosaicism may lead to this effect. We now have mentioned this (lines 409-410).

---

## [Editor Report · Decision Letter 1]

29 Feb 2024

A comparison of software for analysis of rare and common short tandem repeat (STR) variation using human genome sequences from clinical and population-based samples

PONE-D-24-00164R1

Dear Dr. Hollox,

We’re pleased to inform you that your manuscript has been judged scientifically suitable for publication and will be formally accepted for publication once it meets all outstanding technical requirements.

Kind regards,

Paul Aurelian Gagniuc, PhD

Academic Editor

PLOS ONE
---

## [Editor Report · Acceptance letter]

20 Mar 2024

PONE-D-24-00164R1 

PLOS ONE

Dear Dr. Hollox, 

I'm pleased to inform you that your manuscript has been deemed suitable for publication in PLOS ONE. Congratulations! Your manuscript is now being handed over to our production team.

Kind regards, 

on behalf of

Dr. Paul Aurelian Gagniuc 

Academic Editor

PLOS ONE